# Investigating the Effect of Pre-finetuning BERT Models on NLI Involving Presuppositions

**Jad Kabbara**[1]
Massachusetts Institute of Technology
Cambridge, MA, USA
jkabbara@mit.edu

**Jackie Chi Kit Cheung**[2]
McGill University & Mila
Montreal, QC, Canada
jcheung@cs.mcgill.ca

## Abstract

We explore the connection between presupposition, discourse and sarcasm and propose to leverage that connection in a transfer learning scenario with the goal of improving the performance of NLI models on cases involving presupposition. We exploit advances in training transformer-based models that show that pre-finetuning——i.e., finetuning the model on an additional task or dataset before the actual finetuning phase——can help these models, in some cases, achieve a higher performance on a given downstream task. Building on those advances and that aforementioned connection, we propose pre-finetuning NLI models on carefully chosen tasks in an attempt to improve their performance on NLI cases involving presupposition. We notice that, indeed, pre-finetuning on those tasks leads to performance improvements. Furthermore, we run several diagnostic tests to understand whether these gains are merely a byproduct of additional training data. The results show that, while additional training data seems to be helping on its own in some cases, the choice of the tasks plays a role in the performance improvements.

## 1 Introduction

In linguistics and the philosophy of language, presuppositions are beliefs that are mutually shared between discourse participants. They are assumptions that are taken for granted and that play a crucial role in understanding the proper meaning being conveyed in a certain context. For example, in the statement "Roger Federer won the match", the fact that Roger Federer played a match (which he ended up winning) is not explicitly stated and assumed, hence, we say it is *presupposed*. The study of presupposition could be traced to more than a century ago with Frege's work (1892). Given its

importance in understanding meaning and prevalence in everyday speech, scholars studied how presupposition fits into different contexts and aspects of language. Specifically, they investigated how presupposition is intricately related to discourse (Heim, 1992; Kamp and Reyle, 1993) and essential for understanding not only the meaning of an utterance but also essential for the coherence of the semantic relations between sentences forming a discourse (Domaneschi, 2016). Similarly, numerous studies underline how presupposition is linked to sarcasm, dating back to two millennia ago with the work of the Roman rhetorician Quintilian and more recent investigations (Grice, 1975; Camp, 2012).

In this work, we set to exploit this connection between presupposition on one hand and discourse and coherence models and sarcasm on the other hand. We choose a popular NLU task, natural language inference (NLI) (Dagan et al., 2005; MacCartney and Manning, 2008; Bowman et al., 2015) as a testbed for exploring this connection and ask the following question: Could we leverage learning tasks that center around discourse coherence and sarcasm to improve the ability of NLI models at making inferences involving presupposition?

To this end, we propose exploring how such learning tasks can be leveraged in a transfer learning scenario that could benefit the training of NLI models for the final goal of improving their performances on NLI cases involving presupposition. To do so, we utilize pre-finetuning, a training strategy that has received attention in the context of transformer (Vaswani et al., 2017) models such as BERT (Devlin et al., 2019), RoBERTa (Liu et al., 2019), BART (Lewis et al., 2020) and GPT-3 (Brown et al., 2020). Generally, prefinetuning a pretrained model is to finetune it on a task/dataset before the actual finetuning stage on the downstream task of interest. The effect of pre-finetuning varies depending on different factors, such as whether (and to what extent) a pre-finetuning task is relevant to the fine-

---

[1]Most of the work was completed during Jad's PhD at McGill.

[2]Canada CIFAR AI Chair.

tuning task. Pre-finetuning has been explored in the context of question answering (Tafjord et al., 2019), named entities (Shwartz et al., 2020) and broader multi-task learning scenarios (Gururangan et al., 2020; Aghajanyan et al., 2021).

We believe this to be an interesting study for two reasons. First, previous work (Williams et al., 2018; Jeretic et al., 2020) suggested that MNLI (Williams et al., 2018), a popular NLI benchmark, includes some instances of presupposition triggers, thus raising the question of whether transformer-based models trained on MNLI can generalize to other pragmatic inferences. However, previous work (Jeretic et al., 2020; Kabbara and Cheung, 2022) found that BERT does not perform well on many types of presupposition. This underlines how hard it is to design computational models of pragmatic reasoning. In our case, it is thus interesting to explore ways to improve the performance of NLI models on those hard cases of presupposition. Second, given the aforementioned intricate connection between the notions of presupposition, discourse and sarcasm, it would be interesting to investigate whether such tasks—expected to be beneficial in principle—would be useful, in practice, for NLI involving presuppositions.

Beyond the motivating reasons for this specific study, finetuning is still a very popular learning paradigm despite the recent rise of other paradigms like prompt-based methods (Brown et al., 2020) and recent works continue to investigate ways to improve and apply this paradigm (Gira et al., 2022; Zhang et al., 2023; Tinn et al., 2023). Accordingly, we see our effort as complementing the body of work attempting to explore the efficacy and applicability of this learning paradigm.

Our investigation pursues these questions in the context of three pre-finetuning tasks: discourse relation classification, coherence modeling and sarcasm detection. For each of the tasks, we run experiments on two different datasets. We find that, indeed, pre-finetuning on the selected tasks leads to an improvement in the performance of these models on NLI cases involving presupposition given in the ImpPres dataset (Jeretic et al., 2020). Furthermore, we run several diagnostic tests to understand whether these performance improvements are merely a byproduct of additional training data. The results show that, while additional training data seems to be helping on its own in some cases, the choice of the tasks seems to be playing a role in the

performance improvement.

Our contribution is three-fold. First, we draw this connection between presupposition, discourse and sarcasm, showing its effects in neural models which could be beneficial for future investigations in the computational pragmatics area. Second, we present a new case study pointing to a promising direction to train transformer-based models to better handle hard NLI cases involving pragmatic phenomena. Third, through an extensive array of experiments and diagnostic tests, we show that the type of learning tasks used in pre-finetuning, namely discourse-based and pragmatics-based tasks, could play a crucial role in improving the performance of NLI models on cases involving presupposition and show that the performance improvements are not solely due to additional training data, but rather influenced by the learning signal coming from the pre-finetuning task.

## 2 Related Work

### 2.1 Presupposition, Discourse and Sarcasm

The philosophy of language and linguistics literature covered amply the connection between discourse, sarcasm and presupposition. While earlier work on presupposition focused on the pragmatic meaning at the level of utterances (Strawson, 1950; Austin, 1962; Stalnaker, 1973), in recent decades, linguists started looking at presupposition from the angle of dynamic semantic theories and how this pragmatic phenomenon fits within representations of language structure that model the change in information in the course of a discourse (Heim, 1992; Kamp and Reyle, 1993; Potts, 2015). This (re)focus from the level of singular utterances to the level of discourse structure was crucial in highlighting how presuppositions are essential for understanding not only the meaning of an utterance but also essential for the coherence of the semantic relations between sentences forming a discourse (Domaneschi, 2016). In this light, several works presented discourse modeling architectures with the goal of presupposition resolution (Kasper et al., 1999; Davis, 2000). Similarly, the connection between sarcasm and presupposition has been investigated as far back as two millennia ago with the Roman rhetorician Quintilian explaining sarcasm as speech that could be understood as the opposite of what is actually being said. In his authoritative work Logic and Conversation (1975), Grice argues that a speaker speaking sarcastically exploits a pre-

supposition; i.e., a mutually shared assumption, that they could not have plausibly meant what they said. On the other hand, Camp (2012) holds the view that sarcasm does not need to involve a maxim violation (which is usually the case in the Gricean view) and argues that sarcasm is speech that presupposes a normative scale where the speaker pretends to evoke a commitment to this scale but communicates an inversion of that evoked scale. According to Camp, the speaker pretends to mean the inverse of what is being expressed or presupposes someone else to have meant that meaning.

## 2.2 Natural Language Inference

Early work on natural language inference (NLI) such as (Dagan et al., 2005; Manning, 2006; Mac-Cartney and Manning, 2008) framed the task as one concerned with learning the "directional relation" between two statements $t$ and $h$ such that $t$ would entail $h$ if, typically, a human reading $t$ would infer that $h$ is most likely true (Dagan et al., 2005). Contemporary work on NLI (Bowman et al., 2015; Williams et al., 2018) formulates the task as learning the relationship between a premise and a hypothesis such that the relationship is one of entailment, contradiction or neither (in which the two statements are considered to be neutral with respect to each other). Many NLI datasets have been introduced including SNLI (Bowman et al., 2015), MNLI (Williams et al., 2018), MPE (Lai et al., 2017), XNLI (Conneau et al., 2018), SciTail (Poliak et al., 2018), JOCI (Zhang et al., 2017) and others. In our work, we investigate the capabilities of pre-trained transformer-based models finetuned on MNLI (Williams et al., 2018) and their performance on pragmatic inferences in the ImpPres dataset (Jeretic et al., 2020). MNLI is a crowdsourced dataset of 433k sentence pairs that follows the structure of the original SNLI dataset (Bowman et al., 2015) but is richer as it covers a range of genres of spoken and written text and is tailored towards a cross-genre generalization evaluation. The ImpPres dataset is a collection of 25.5k semi-automatically generated sentences also structured following SNLI/MNLI with the goal of evaluating how well NLI-trained models recognize several classes of presupposition and implicature. In this work, we focus on the presupposition part of ImpPres and show in Table 1 an example from each sub-dataset to make the later discussions clearer. By construction, each of the 9 sub-datasets (target-

ing the different types of presupposition shown in Table 1) contains 1900 samples.

## 2.3 Pre-finetuning Transformer-based Learning Models

In (Tafjord et al., 2019), the authors found that an additional pre-finetuning stage on a different multiple choice question (MCQ) dataset improves the model's generalization to another MCQ dataset, suggesting that the pre-finetuning stage seems to have helped the model learn some representations about MCQs in general. In (Shwartz et al., 2020) which investigates the robustness of language models to name swaps, it was found that a pre-finetuning stage on a different dataset (in some cases pertaining to a different NLP task) led to small performance improvements on the main task of interest (Winogrande (Sakaguchi et al., 2020) or SQUAD (Rajpurkar et al., 2018)) but seemed to increase the model's robustness with respect to name swaps. Gururangan et al. (2020) consider different domains and classification tasks and show that an intermediate stage (between pre-training and finetuning) of further pre-training on a relevant task leads to a performance boost for their model (RoBERTa). Finally, Aghajanyan et al. (2021) present an extensive study on the benefits of pre-finetuning with 50 different tasks. They show that pre-finetuning consistently improves performance for pre-trained discriminators (e.g. RoBERTa) and generation models (e.g. BART) on a wide range of tasks (sentence prediction, commonsense reasoning, machine reading comprehension, etc.), while also significantly improving sample efficiency during fine-tuning. Their experiments show that a small number of pre-finetuning tasks does not always help but that for 15+ tasks (up to 50), the improvements become consistent and linear in the number of tasks. Given that pre-finetuning is not always helpful (Aghajanyan et al., 2021), one contribution of this work is to highlight that a carefully chosen task that exploits the connection between presupposition, discourse and sarcasm could improve the performance of NLI models on NLI cases involving presupposition. Other works, e.g., (Pruksachatkun et al., 2020; Poth et al., 2021), while not specifically focused on the notion of pre-finetuning, also showed that further (pre-)training on a carefully chosen can also improve performance on downstream tasks.

| Type | Premise | Hypothesis |
|------|---------|------------|
| All N | All six roses that bloomed died. | Exactly six roses bloomed. |
| Both | Both flowers that bloomed died. | Exactly two flowers bloomed. |
| Change of State | Rene might have hidden. | Rene hid. |
| Cleft Existence | It might be Becky who researched Jesus. | Someone researched Jesus. |
| Cleft Uniqueness | It is Sandra who disliked Veronica. | Exactly one person disliked Veronica. |
| Only | Susan only writes. | Susan writes. |
| Possessed Definites | Alice's light did vanish. | Alice has a light. |
| Question | Did Bill wonder when Omar hunted? | Omar hunted. |

Table 1: Examples showing the different presupposition types in the ImpPres dataset.

## 3 Proposed Method

The learning framework consists of using a pre-trained transformer-based model to be finetuned on the MNLI dataset. This last stage is preceded by a pre-finetuning stage in which the model is trained on another task different than NLI. The model is finally evaluated on the ImpPres presupposition dataset whose goal is to test NLI models' ability to recognize different classes of presupposition. Given that the final evaluation is concerned with the model's ability to make pragmatic inferences, we believe an interesting case study would be to use pre-finetuning tasks that deal with aspects of discourse and pragmatics. That way, we could attempt to understand whether relevant knowledge learned from the pre-finetuning stage is passed down to the latter stages.

### 3.1 Pre-finetuning Tasks and Datasets

#### 3.1.1 Discourse Relation Classification

Discourse relation classification (DRC) is a commonly used task for evaluating the understanding of discourse relations. The task focuses on characterizing the relation between two adjacent text spans (either clauses or sentences) which could be related by an explicit or implicit relation. In our work, we focus on the top-level (L1) classification scenario (according to the PDTB 2.0 structure (Prasad et al., 2008)) in which a discourse relation takes one of 4 labels: Comparison, Contingency, Expansion and Temporal. We use two datasets for this task: PDTB3.0 (Prasad et al., 2019) and the TED Multilingual Discourse Bank (TED-MDB) (Zeyrek et al., 2019) which is a PDTB-style multilingual dataset consisting of TED-talks that are annotated at the discourse level in 6 languages. We use in our work the English portion of the dataset only.

#### 3.1.2 Closed-Domain Coherence Modeling

We focus here on another discourse-based task: Coherence Modeling (CM), the task concerned with classifying a passage as being coherent or incoherent. The most popular task used to test a coherence model in NLP is sentence ordering[3], for example, to distinguish between a coherently ordered list of sentences and a random permutation thereof. In the closed-domain setup (Tien Nguyen and Joty, 2017), training and testing are done on the same domain. Similar to previous work (Tien Nguyen and Joty, 2017; Xu et al., 2019), we use use the WSJ portion of the PTB dataset (Marcus et al., 1993). We also use the Wiki-A dataset (Xu et al., 2019) (which uses one domain/topic "Person" and splits all the different categories therein randomly into training/testing parts).

#### 3.1.3 Sarcasm Detection

In the last task, we focus on a task whose successful execution requires an understanding of the underlying pragmatics conveyed in the text or conversation. The task is sarcasm detection (SD) in which a model is to classify a text as being sarcastic or not. For this task, we use the following two commonly used datasets: The Irony and Sarcasm dataset (Filatova, 2012) which is a collection of Amazon reviews annotated as being sarcastic/ironic or not, and the News Headlines Dataset (Misra and Arora, 2019) which is a dataset consisting of headlines of articles from the Huffington Post (labeled as non-sarcastic) and from the satirical online media outlet, The Onion (labeled as sarcastic).

---

[3]We use this task as it is still the most popular for the evaluation of coherence models. However, we note that some recent work, e.g., (O'Connor and Andreas, 2021; Laban et al., 2021; Jeon and Strube, 2022) has pointed out the limitations of this task in effectively evaluating coherence models (e.g., simplicity, reliance on local word co-occurrences but not word order). And so, for an investigation that is solely focused on coherence modeling, it is advised to also use other tasks for evaluation purposes. See the cited work for suggestions.

## 4 Experiments

In this work, we attempt to answer whether pre-finetuning an NLI model on learning tasks that deal with aspects of discourse and pragmatics would help that model, when finetuned on MNLI, perform better on pragmatic NLI cases (the ImpPres dataset). To do so, we carry out three different experiments.

1. The main setup (MAIN): We pre-finetune the model on each dataset of the three learning tasks (6 in total - Section 3). Following finetuning on MNLI, the model is evaluated on the ImpPres dataset.

2. Randomizing the labels (RANDOM): To understand whether any performance improvements are due to extra training data or some actual learning signal coming from the pre-finetuning stage, we randomly shuffle the labels (using a uniform distribution) in each of the datasets used in the pre-finetuning stage. The rationale behind this experiment is to see whether, given corrupted data, the performance would drop. This was can be thought of as a "sanity check".

3. Using the datasets for further pre-training (PRETRAIN): Here, we use the input samples from each dataset to further pre-train the BERT model using a Masked Language Modeling objective. This is to further understand whether extra pre-training would lead to similar changes in the performance (thus possibly suggesting that a perfomance gain is due to extra training data and not to the learning tasks used in the pre-finetuning stage).

Finally, we also run an experiment (MULTI) where we pre-finetune the model on all three tasks, one after the other. Due to limited computational resources, we do not run all possible combinations of datasets. We chose one order at random to be presented as a proof of concept. The chosen sequence of tasks/datasets is: DRC/PDTB then CM/WSJ then SD/Reviews.

### 4.1 Implementation details

The main building block of our experiments is HuggingFace's bert-large-uncased implementation (Wolf et al., 2019) of BERT that was trained on lower-cased English text.

Depending on the pre-finetuning task, the base BERT model is followed by a corresponding linear layer and softmax for classification (2-way or 4-way classification depending on the task) or none in the case of the PRETRAIN experiment (where the data is used to further pre-train the model and not to predict labels). For the classification-based pre-finetuning stages and the MNLI finetuning stage (common to all experiments), the model is trained to minimize the standard cross-entropy cost with Adam (Kingma and Ba, 2015) as the optimizer. For the PRETRAIN experiments, in the pre-finetuning stage, the model is (further pre-)trained using a masked language modeling objective. For the MNLI finetuning stage, similar to previous work using BERT for NLI, we concatenate the premise and hypothesis separated by the [SEP] token, with the special [CLS] token preceding them.

Following the recommended ranges for finetuning hyperparameters in the BERT paper (Devlin et al., 2019), our preliminary experiments showed that, for finetuning BERT only (i.e. no pre-finetuning), the optimal performance on the dev set is reached for a batch size of 8, learning rate of 2e-5 and weight decay of 0.01. Our model achieves a dev set accuracy of 85.14% (comparable to that reported in (Jeretic et al., 2020) and (Devlin et al., 2019). We fix those hyperparameters for the finetuning stage across all experiments and vary the learning rate and weight decay for the pre-finetuning stage across the following ranges respectively: {1e-5, 2e-5, 3e-5, 4e-5} and {0.1, 0.01, 0.001}.[4] All other parameters are kept as default. All models are implemented in PyTorch (Paszke et al., 2019). All input data is tokenized by HuggingFace's BERT tokenizer (bert-large-uncased). All experiments were run on a Quadro RTX 6000 GPU.

## 5 Results

Table 2 shows the accuracy results of the three experiments MAIN, RANDOM and PRETRAIN.

**MAIN:** In the first sub-table, we notice that the addition of a pre-finetuning stage leads to an improvement in accuracy in the vast majority of cases across pre-finetuning tasks/datasets through the different sub-datasets of ImpPres–at least 6 out of the 9 sub-datasets of ImpPres for all tasks/datasets and

---

[4]We include in the appendix the exact learning rate and weight decay found for optimal performance on dev set for the pre-finetuning stage for each of the experiments.

| MAIN | Vanilla | Discourse Relation Classification | | Coherence Modeling | | Sarcasm Detection | |
|---|---|---|---|---|---|---|---|
| | | PDTB | TED-MDB | WSJ | Wiki | Reviews | Headlines |
| Possessed definites existence | 70.61 | **71.85** | **71.78** | **72.06** | **71.27** | **71.32** | **71.48** |
| Question | 66.42 | 65.23 | 64.90 | **67.33** | **67.33** | 65.18 | 63.34 |
| Cleft Existence | 62.97 | 62.13 | 61.97 | 61.55 | **65.44** | 61.66 | 60.24 |
| Only | 62.24 | **62.24** | 61.26 | 60.56 | **62.76** | **62.87** | 60.03 |
| All n | 43.49 | **49.21** | **46.69** | **44.85** | **44.59** | **45.80** | **46.01** |
| Both | 32.62 | **34.30** | **33.91** | **33.25** | 24.32 | 27.78 | **38.66** |
| Change of state | 30.43 | **31.93** | **33.63** | **32.62** | **34.19** | **31.57** | **32.62** |
| Possessed definites uniqueness | 23.35 | **33.04** | **25.33** | **35.71** | **30.36** | **31.57** | **33.98** |
| Cleft uniqueness | 11.10 | **14.60** | **16.51** | **20.90** | **16.91** | **13.24** | **13.92** |
| Average | 44.80 | **47.17** | **46.22** | **47.42** | **46.35** | **45.67** | **46.70** |

| RANDOM | Vanilla | Discourse Relation Classification | | Coherence Modeling | | Sarcasm Detection | |
|---|---|---|---|---|---|---|---|
| | | PDTB | TED-MDB | WSJ | Wiki | Reviews | Headlines |
| Possessed definites existence | 70.61 | 61.71 | 68.33 | 68.96 | 66.86 | 69.17 | 69.07 |
| Question | 66.42 | 62.71 | 63.92 | 63.81 | 63.66 | 64.97 | 65.39 |
| Cleft Existence | 62.97 | 60.56 | 60.77 | 60.61 | 60.14 | 60.87 | 60.82 |
| Only | 62.24 | 61.03 | 55.46 | 60.29 | 56.51 | 60.98 | 60.98 |
| All n | 43.49 | **44.91** | 42.12 | 38.76 | **43.80** | 41.91 | **43.80** |
| Both | 32.62 | 25.16 | 22.48 | 22.95 | 27.63 | 24.63 | 26.58 |
| Change of state | 30.43 | 26.21 | 27.21 | **31.25** | 28.36 | **31.36** | 29.57 |
| Possessed definites uniqueness | 23.35 | **24.74** | **34.09** | 23.21 | 18.59 | **25.95** | **27.21** |
| Cleft uniqueness | 11.10 | **12.39** | **14.71** | **14.02** | 10.77 | **12.18** | **12.50** |
| Average | 44.80 | 42.16 | 43.23 | 42.65 | 41.81 | 43.56 | 43.99 |

| PRETRAIN | Vanilla | Discourse Relation Classification | | Coherence Modeling | | Sarcasm Detection | |
|---|---|---|---|---|---|---|---|
| | | PDTB | TED-MDB | WSJ | Wiki | Reviews | Headlines |
| Possessed definites existence | 70.61 | 69.36 | 66.6 | **72.87** | **71.94** | 65.96 | 70.16 |
| Question | 66.42 | 64.48 | 65.97 | 65.46 | 65.63 | 62.91 | 64.64 |
| Cleft Existence | 62.97 | 62.76 | 58.77 | 61.33 | 61.92 | 62.07 | **63.33** |
| Only | 62.24 | **63.39** | **62.87** | 60.49 | 59.72 | 57.39 | 59.70 |
| All n | 43.49 | **43.80** | 40.91 | 40.04 | 42.02 | 38.41 | 42.56 |
| Both | 32.62 | 29.46 | 30.93 | 30.26 | 24.68 | 20.48 | 27.37 |
| Change of state | 30.43 | **31.88** | **32.35** | **31.02** | 30.57 | 29.84 | 28.84 |
| Possessed definites uniqueness | 23.35 | 20.41 | 11.45 | **24.79** | **35.59** | 11.04 | **39.14** |
| Cleft uniqueness | 11.10 | **14.44** | **13.39** | 10.17 | **12.55** | 9.70 | **15.27** |
| Average | 44.80 | 44.44 | 42.58 | 44.04 | **44.96** | 39.76 | **45.67** |

Table 2: Results showing accuracy performance on the various sub-datasets of ImpPres for the three experiments: MAIN, RANDOM and PRETRAIN . Vanilla refers to no pre-finetuning. For each column, we emphasize in bold the cases where the accuracy is higher than the corresponding one in the Vanilla case. As a reminder, all sub-datasets have the same number.

|                                   | Vanilla | Multi   |
|-----------------------------------|---------|---------|
| Possessed definites existence     | 70.61   | **71.36** |
| Question                          | 66.42   | **67.58** |
| Cleft Existence                   | 62.97   | **65.34** |
| Only                              | 62.24   | **63.15** |
| All n                             | **43.49** | 42.03   |
| Both                              | **32.62** | 25.11   |
| Change of state                   | 30.43   | **35.77** |
| Possessed definites uniqueness    | 23.35   | **27.38** |
| Cleft uniqueness                  | 11.10   | **14.83** |
|                                   | 44.80   | **45.84** |

Table 3: Accuracy results for the MULTI experiment.

going up to 8/9 sub-datasets in the case of the Coherence Modeling task with the Wiki dataset. Moreover, in all 6 scenarios (pre-finetuning task/dataset), we notice that there is an improvement in the average accuracy performance compared to the average performance in the Vanilla case. This suggests that a pre-finetuning stage using suitable tasks/datasets could help the performance of MNLI-trained models on pragmatic NLI cases.

**RANDOM:** To further validate that hypothesis, we notice in the RANDOM experiment (second subtable) that when we randomize the labels the average performance drops in all 6 scenarios compared to the Vanilla scenario. Moreover, the performance drops in the vast majority of cases (pre-finetuning task/dataset × ImpPres sub-dataset).

**PRETRAIN:** On the other hand, when we use the training data from the pre-finetuning stage for further pre-training (the PRETRAIN experiment) instead of the corresponding learning tasks, we notice that the average performance drops in 4 of the 6 scenarios (pre-finetuning task/dataset) along with a drop in most individual cases (pre-finetuning task/dataset × ImpPres sub-dataset). This suggests that, in certain cases, the use of extra pre-training data could help in boosting the performance. However, the performance gains due to further pre-training is far from consistent compared to what we saw as a result of using a pre-finetuning stage (i.e. the MAIN experiment).

**MULTI:** Table 3 shows the accuracy results obtained when using three simultaneous tasks in the pre-finetuning stage. We notice that using a pre-finetuning stage with 3 tasks leads to an increase in the accuracy performance in 7 out of 9 sub-datasets as well as an improvement in the average accuracy

performance. Comparing these results to those of the MAIN experiment, we observe that using 3 simultaneous tasks is not necessarily beneficial as the average accuracy performance in this case is lower than most scenarios seen in the MAIN experiment.

## 5.1 Discussion

While the margin of improvement is mostly small between a given case (pre-finetuning task/dataset × ImpPres sub-dataset) and the corresponding vanilla case, we note this margin is comparable to what is reported in related studies using pre-finetuning (e.g., (Aghajanyan et al., 2021; Shwartz et al., 2020)). Moreover, we notice that for 4 of the 9 sub-datasets (namely, the All n, Change of state, Possessed definites uniqueness and Cleft uniqueness datasets), there is full consistency in terms of performance improvement across all pre-finetuning tasks and corresponding datasets (that is, for these 4 sub-datasets, pre-finetuned models outperform the vanilla model in all tasks/corresponding datasets).

The results from the MAIN experiment on their own are quite interesting because previous work (Aghajanyan et al., 2021) that ran an extensive empirical study on 50 different pre-finetuning tasks showed that using a small number of pre-finetuning tasks does not necessarily lead to a performance gain on downstream NLI-based tasks and, in fact, could sometimes hurt the performance. The results presented here suggest that a *single* suitable pre-finetuning task could indeed help MNLI-trained models perform better on pragmatic NLI cases.

Beyond accuracy improvements, what is even more interesting is what the two diagnostic tests RANDOM and PRETRAIN–combined with the accuracy improvements seen in the MAIN experiment– seem to be suggesting. The suggestion is that the choice of the learning task in the pre-finetuning stage is playing a role in the performance improvements and that those improvements are in most cases not a result of simply extra training data. Moreover, given that these pre-finetuning tasks are heavily based on aspects of discourse and pragmatics, the nature of these tasks is relevant to the pragmatic nature of the NLI cases that make up the ImpPres dataset. We thus believe that the performance improvement is due in large part to cross-task knowledge transfer from the pre-finetuning stage to the later stages: This is because the learning model was exposed to little focus on notions of pragmatics when it was (originally) only finetuned

|  | Vanilla | POS | NER |
|---|---|---|---|
| Poss. def. existence | 70.61 | 70.17 | 58.93 |
| Question | 66.42 | 61.34 | 63.87 |
| Cleft Existence | 62.97 | 60.77 | **64.71** |
| Only | 62.24 | 59.19 | 58.51 |
| All n | 43.49 | 39.55 | 32.77 |
| Both | 32.62 | 23.48 | 27.84 |
| Change of state | 30.43 | **31.14** | **32.04** |
| Poss. def. uniqueness | 23.35 | 22.15 | 18.80 |
| Cleft uniqueness | 11.10 | **15.70** | **18.99** |
|  | 44.80 | 42.61 | 41.83 |

Table 4: Accuracy results for the experiments involving pre-finetuning on POS tagging and NER.

on MNLI and was now being made richer through the aforementioned cross-task knowledge transfer.

Regarding the MULTI experiment, the fact that the performance of the model pre-finetuned on 3 simultaneous tasks is comparable or lower in some cases is in line with previous work (Aghajanyan et al., 2021) that found that performance improvements become linear with the number of tasks only starting from 15 pre-finetuning tasks. Also, as we already mentioned, due to computational resources, the hyperparameters (learning rate, weight decay) were fixed for all 3 pre-finetuning tasks and not optimized for each task on its own which could've been sub-optimal thus affecting the performance.

### 5.2 Pre-finetuning on other tasks

One question that might arise is whether other pre-finetuning tasks that are not directly related to notions of discourse or pragmatics would help in a similar fashion to what we've seen in the main experiments (Section 4). To further explore this, we carry out two experiments with different pre-finetuning tasks: POS tagging and Named Entity Recognition (NER). For the first experiment, we use WSJ-PTB (Marcus et al., 1993) with the standard section splits. For the second experiment, we use the English portion of the CoNLL-2003 NER dataset (Tjong Kim Sang and De Meulder, 2003). We follow the same training setup as in Section 4.1. Results in Table 4 show a drop in overall average performance for both experiments along with a drop in performance for the vast majority of ImpPres sub-datasets, further strengthening the hypothesis that pre-finetuning on tasks specifically centering on discourse and pragmatics leads to

|  | PDTB | | Wiki | | Reviews | |
|---|---|---|---|---|---|---|
|  | No-P | W/P | No-P | W/P | No-P | W/P |
| neutral | 0 | 55 | 0 | 58 | 0 | 52 |
| entailment | 46 | 1 | 56 | 0 | 47 | 1 |
| contradiction | 10 | 0 | 2 | 0 | 6 | 0 |

Table 5: Distribution of predictions for the Possessed Definites Existence sub-dataset for the samples that were incorrectly predicted by the Vanilla model (no pre-finetuning - No-P) but correctly predicted by the pre-finetuned model (W/P). The No-P column shows the distribution of predictions that were incorrectly made by the Vanilla model (No-P) and how their distribution became when the predictions (on the same samples) were correctly made by the pre-finetuned model (W/P).

an improvement in the performance of transfomer models on NLI cases involving presupposition.

## 6 Sample Analysis

To understand better the effect of the pre-finetuning stage, we analyze the outputs of the model with and without pre-finetuning. Specifically, we look at the models that are pre-finetuned on PDTB (Discourse Relation Classification), Wiki (Coherence Modeling) and the Reviews dataset (Sarcasm Detection). As a case study, we choose the Possessed Definites Existence sub-dataset of ImpPres as the models performance is the highest on this sub-dataset. We investigate the samples that were incorrectly predicted by the Vanilla model (no pre-finetuning) but correctly predicted by the pre-finetuned model for each of these 3 cases. In all 3 cases, there were less than 60 such samples. Table 5 shows the results.

In all 3 pre-finetuning tasks, we notice that, in the vast majority of samples (>95%), the correct label is neutral. While the ImpPres dataset is structured such that the neutral cases are generally easier than the other cases (e.g. the contradiction cases are generally the hardest), the model still got these cases wrong, instead predicting them as entailment in most cases and contradiction in others. However, with pre-finetuning, the models correctly predicted those cases as neutral instead. For example, in cases predicted incorrectly as contradiction, we notice the presence of negation in the premise. The model might have been using this (incorrectly) as an indicator for contradiction. One example is the following: The premise is "That actor's politics didn't upset Nancy", the hypothesis is "Kenneth has politics." The Vanilla model predicted this sample to be one of contradiction whereas the pre-finetuned model predicted it correctly as neutral.

# 7   Conclusion

Pre-finetuning transformer-based models has been explored in recent literature as a means to improve the performance of these models on various tasks. In our work, we investigate the use of a pre-finetuning step in attempt to improve the performance of MNLI-trained models on pragmatic NLI cases. Particularly, we examine three pre-finetuning tasks: discourse relation classification, coherence modeling and sarcasm detection. We focus on these tasks because they are discourse-based or require some notion of pragmatic understanding of the text. We notice that, indeed, pre-finetuning on the selected tasks leads to an improvement in the performance of these models on NLI cases involving presupposition. Furthermore, we run several diagnostic tests to understand whether these performance improvements are merely a byproduct of additional training data. The results show that, while additional training data seems to be helping on its own in some cases, the choice of the tasks seems to be playing a role in the performance improvement. In future work, we intend to investigate other pre-finetuning tasks for the final goal of evaluating NLI models on NLI cases involving other types of pragmatic phenomena (e.g., implicature). Moreover, we are interested in exploring different learning strategies for making transformer-based models better at pragmatic NLI cases.

# 8   Limitations

Evidently, one limitation of this work is that results are shown for the English language only. In principle, we expect that cross-task knowledge transfer from tasks involving notions of discourse and pragmatics to help in the performance of learned models on downstream tasks that also involve notions of pragmatics (as is the case in this work). However, to what other languages this generalizes to (and to what extent)–especially languages that are morphologically richer than English–remains a question for further exploration. Another limitation is that the ImpPres dataset which is the focus of evaluation in our work is a dataset of semi-automatically generated sentences. The generation process involves pre-specified templates that are to be filled with constituents sampled from a limited vocabulary (3000 lexical items). Thus a large-scale quality check process would be required to validate the quality of the entirety of the dataset (e.g., it is mentioned in (Jeretic et al., 2020) that the generated

sentences often describe highly unlikely scenarios or might include combinations of lexical items that make the sentence sound unnatural).

## Acknowledgements

The authors would like to thank the reviewers for their valuable comments. Jad Kabbara was supported by an IVADO PhD fellowship. This work is partially supported by the Canada CIFAR AI Chair Program held by Jackie Chi Kit Cheung at Mila.

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

# A   Appendix

## A.1   Additional training details

We present here the learning rate and weight decay that were obtained for each model following hyperparameter search as explained in Section 4.1.

| MAIN | Discourse Relation Classification | | Coherence Modeling | | Sarcasm Detection | |
|---|---|---|---|---|---|---|
| | PDTB | TED-MDB | WSJ | Wiki | Reviews | Headlines |
| Weight decay | 0.1 | 0.1 | 0.001 | 0.1 | 0.1 | 0.01 |
| Learning rate | 2e-5 | 4e-5 | 2e-5 | 1e-5 | 4e-5 | 2e-5 |

| RANDOM | Discourse Relation Classification | | Coherence Modeling | | Sarcasm Detection | |
|---|---|---|---|---|---|---|
| | PDTB | TED-MDB | WSJ | Wiki | Reviews | Headlines |
| Weight decay | 0.001 | 0.1 | 0.1 | 0.1 | 0.001 | 0.1 |
| Learning rate | 1e-5 | 2e-5 | 2e-5 | 1e-5 | 2e-5 | 3e-5 |

| PRETRAIN | Discourse Relation Classification | | Coherence Modeling | | Sarcasm Detection | |
|---|---|---|---|---|---|---|
| | PDTB | TED-MDB | WSJ | Wiki | Reviews | Headlines |
| Weight decay | 0.001 | 0.01 | 0.001 | 0.1 | 0.1 | 0.01 |
| Learning rate | 4e-5 | 4e-5 | 3e-5 | 1e-5 | 1e-5 | 4e-5 |