# OpenReview forum: "Investigating the Effect of Pre-finetuning BERT Models on NLI Involving Presuppositions"
_EMNLP/2023/Conference — EMNLP 2023 Findings_

### Official Review · Reviewer_joCE · 2023-08-04

**Soundness:** 4

**Excitement:**

4: Strong: This paper deepens the understanding of some phenomenon or lowers the barriers to an existing research direction.

**Paper Topic And Main Contributions:**

This paper investigates the method of pre-finetuning, which involves finetuning a model on an additional task before fine-tuning it on the target task. Specifically, the authors pre-finetune on 3 tasks (2 datasets each): discourse relation classification, coherence modelling, and sarcasm detection, all of which involve the linguistic phenomenon of presupposition. They then investigate whether this pre-finetuning stage is beneficial for the target task, namely the ImPres dataset, which also involves presupposition.

The paper makes the following contributions:

- Continues research on an element of discourse (presupposition) that is easy for humans, but still challenging for models.
- Sheds light on the benefits of pre-finetuning, by showing that it leads to performance gains on the target task given related pre-finetuning tasks.
- Continues research on fine-tuning, which remains an important method despite the current focus on prompting.

**Questions For The Authors:**

A) Why not experiment with additional architectures, e.g. RoBERTa which generally outperforms BERT?

**Reasons To Accept:**

- To increase models' understanding of language, it's important to experiment on elements of discourse that humans find easy, but models find difficult. In particular, it's important to investigate how we can make these elements useful to a model. This paper sheds light on one such element of discourse, namely presupposition. This paper shows how models can make use of pre-finetuning signal containing presupposition to solve target tasks that involve presupposition in some way. Additionally, the paper provides a very good linguistics-focused background and explanation of presupposition in the Related Works section, making the reader understand the significance of presupposition for humans, as well as underscoring the importance of including it in NLP analysis.

- The paper makes the important finding that pre-finetuning on tasks that involve presupposition does improve performance on a target task that involves presupposition. This provides an encouraging foundation for future research.

- The authors perform a model output analysis starting from Line 541, which provides a helpful glimpse into the models' exact behavior. I believe this is an important practice, since models are generally becoming less interpretable.

**Reasons To Reject:**

- Line 355 ("Randomizing the labels"): The authors' intention to test whether performance gains are merely due to extra training data is logical, but to me, it seems like corrupting the data isn't a convincing way to do this. Corrupted data plays the role of noise, which generally tends to be harmful to performance. So one could argue that the low scores observed in the "Random" setting in Table 2 were due to label noise. Additionally, even if the performance had remained relatively the same, this would have said more about the model's robustness against noise than about whether or not the mere addition of more data originally helped it perform well. I think a better way to conduct the intended test would be to perform some form of over- and under-sampling on the data, such as duplication, and using that for pre-finetuning. This would ensure that the only thing that's being changed during the test is data quantity, and not label quality. Otherwise, I think a test should have been done to demonstrate that the impact of label noise was minimal on the "Random" setting.

- Additionally, it's not stated whether the results in Tables 2, 3, and 4 are from single runs or averages of multiple runs. Therefore it could be the case that the method isn't robust.

**Reproducibility:**

4: Could mostly reproduce the results, but there may be some variation because of sample variance or minor variations in their interpretation of the protocol or method.

**Reviewer Confidence:**

3: Pretty sure, but there's a chance I missed something. Although I have a good feel for this area in general, I did not carefully check the paper's details, e.g., the math, experimental design, or novelty.

**Typos Grammar Style And Presentation Improvements:**

- For the sample analysis in Section 6, it would have been helpful to have a small table showcasing some of the the responses of models with and without pre-finetuning.

- In the caption of Table 2, I would state that the ImPres dataset was used as the test set as an extra reminder for the reader. Otherwise, the test set seemed to get lost in the other details of the paper.

- Table 2, row "Cleft Uniqueness", column "Vanilla": the value is 11.10 for "Main", but 11.1 for "Random" and "Pretrain". For consistency, I'd change those to 11.10.

---

> ### Author Rebuttal · Authors · 2023-08-28
>
> We would like to thank Reviewer joCE for their encouraging feedback and positive comments and for mentioning the many things they liked about our paper. Specifically, we appreciate that the reviewer found that our paper presents a very good linguistics-focused background and explanation of presupposition, in a way that makes the reader understand the significance of presupposition for humans, as well as a way that underscores the importance of including it in NLP analysis. We thank the reviewer for their view that our paper presents an important finding regarding pre-finetuning on the chosen tasks. Finally, we appreciate the reviewer’s feedback that our analysis  presents an important practice given the increasing difficulty in evaluating model interpretability.
>
> Thank you for your comment and suggestions regarding the label randomization experiment. We agree with all points raised, specifically that introducing noise is expected to lower the model performance and in the case where the performance remained stable, this would’ve shed light on the robustness of the model. Our motivation behind this experiment was for it to be more of a sanity check because sometimes models could behave in unexpected ways. We wanted to make sure that the performance would not actually improve because this would’ve necessitated further inquiry. The experiments you suggest are definitely insightful and we can add a comment that these are other tests that could be carried out.
>
> Regarding other architectures like RoBERTa, the main limitation for us is the computational resources. Given the number of datasets and setups and the resulting number of experiments, we had to limit our experiments to BERT.
>
> We would like to thank the reviewer for the style and presentation improvement suggestions. We’d be happy to update these and incorporate them in the camera-ready version.

---

### Official Review · Reviewer_Gb86 · 2023-08-05

**Soundness:** 3

**Excitement:**

4: Strong: This paper deepens the understanding of some phenomenon or lowers the barriers to an existing research direction.

**Paper Topic And Main Contributions:**

The paper targets a narrow phenomenon (presupposition) in the context of improving natural language inference (NLI) models.

The premise of the paper is that pre-finetuning models that will be later fine-tuned for NLI improve their performance. Moreover, if targeting a specific subproblem in NLI one may be able to improve model performance on teh target subproblem by pre-finetuning on an adequate task.



**Questions For The Authors:**

It is unclear to what extent the phenomenon targeted (presupposition) is one that is frequently occurring in practice when using NLI models, and so to what extent improvements on this subproblem are impactful.

**Reasons To Accept:**

NLI is a foundational component in evaluating generative text models and so any significant improvement in this area is very likely to have wide impact.

**Reasons To Reject:**

The wealth of experimental data and really narrow scope of the paper risks to make it relevant to a very narrow audience.

**Reproducibility:**

4: Could mostly reproduce the results, but there may be some variation because of sample variance or minor variations in their interpretation of the protocol or method.

**Reviewer Confidence:**

2: Willing to defend my evaluation, but it is fairly likely that I missed some details, didn't understand some central points, or can't be sure about the novelty of the work.

**Typos Grammar Style And Presentation Improvements:**

It would be good if the paper was accessible to the non-expert in NLI. A summary of results and their significance to the NLI field itself would be good.

---

> ### Author Rebuttal · Authors · 2023-08-28
>
> We thank Reviewer Gb86 for their feedback and comments. We appreciate the reviewer's strong excitement towards our paper and appreciate that the reviewer believes that our work focuses on a foundational problem in evaluating generative text models.
>
> Thank you for your comments regarding the scope and impact of the work. Below we’d like to respond to these comments.
>
> While we focus on presupposition in this work, we believe that our work, first and foremost, highlights an angle on NLI that is under-explored in the literature, specifically NLI involving pragmatics. Moreover, as you rightly point out, we also believe that NLI is a foundational benchmark for evaluating the progress of NLP models on understanding and our work targets a hard problem in the text understanding space, namely pragmatic understanding. Accordingly, we believe our research sheds light on a new and unexplored angle in the NLI space and has the potential to encourage new research on assessing and understanding capabilities of NLP models. This is especially timely with the renewed interest in language models and the recurring questions about the extent to which they exhibit understanding given their impressive performance on many tasks. We also believe that our work can encourage research that explores other kinds of pragmatic phenomena and their intersection with the NLU space and our experiments could inspire future research to showcase more rigor when evaluating NLP models.

---

### Official Review · Reviewer_PE3F · 2023-08-05

**Soundness:** 4

**Excitement:**

2: Mediocre: This paper makes marginal contributions (vs non-contemporaneous work), so I would rather not see it in the conference.

**Missing References:**

[1] Yada Pruksachatkun, Jason Phang, Haokun Liu, Phu Mon Htut, Xiaoyi Zhang, Richard Yuanzhe Pang, Clara Vania, Katharina Kann, and Samuel R. Bowman. Intermediate-Task Transfer Learning with Pretrained Language Models: When and Why Does It Work?. ACL 2020.

[2] Clifton Poth, Jonas Pfeiffer, Andreas Rücklé, and Iryna Gurevych. What to Pre-Train on? Efficient Intermediate Task Selection. EMNLP 2021.


**Paper Topic And Main Contributions:**

This paper presents a study on whether pre-finetuning NLI models on intermediate tasks related to discourse and pragmatics would improve the models’ performance on NLI cases involving presupposition.

The pre-finetuning datasets are chosen from three tasks, including discourse relation classification, coherence modeling, and sarcasm detection. Experiments on the BERT model show that all pre-finetuning datasets lead to improved average performance on 9 subsets of ImpPres, an NLI datasets focused on presupposition and implicature.

Further ablation studies show that the improved performance is not merely a byproduct of additional training data, as the choice of the tasks also plays an important role in the performance boost.

Main contributions:

a) Empirically established the connection between presupposition, discourse and pragmatics by showing the performance improvement from pre-finetuning.

b) A case study that might inspire similar future work on improving transformer-based models’ performance on other hard tasks.


**Questions For The Authors:**

a) While pre-finetuning on tasks related to discourse and pragmatics improves performance on ImpPres, does it also impact the model’s performance on MNLI? It would be helpful to include some analysis to see whether the performance improvement from pre-finetuning also comes at a cost.


**Reasons To Accept:**

a) The paper is well written and easy to follow

b) The experiments are well-motivated with extensive ablation studies


**Reasons To Reject:**

a) There are many previous works (for example, [1] [2] in missing references below) on pre-finetuning with intermediate tasks that are not discussed in this paper.

b) In particular, one of the main claims in the paper that one single carefully chose task can be beneficial in pre-finetuning has already been studied and proven in previous work ([1] [2]), where they also experimented with a large number of different combinations of intermediate tasks and target tasks, making this paper less exciting.


**Reproducibility:**

4: Could mostly reproduce the results, but there may be some variation because of sample variance or minor variations in their interpretation of the protocol or method.

**Reviewer Confidence:**

4: Quite sure. I tried to check the important points carefully. It's unlikely, though conceivable, that I missed something that should affect my ratings.

---

> ### Author Rebuttal · Authors · 2023-08-28
>
> We thank Reviewer PE3F for their encouraging feedback and comments and for highlighting the many things that they liked about our paper. We thank the reviewer for their comment that the paper is well written and easy to follow and that the experiments are well-motivated and that we present extensive ablation studies. We appreciate the reviewer’s feedback that our work can inspire similar future work on improving transformer-based models’ performance on other hard tasks.
>
> Below we'd like to respond to the main points:
>
> 1. Thank you for bringing up the two references [1] and [2]. While both references offer valuable insights and present a wide array of experiments, our work is different in the sense that is focused on NLI and its intersection with pragmatics and to what extent NLI models exhibit some sort of pragmatic competence (specifically related to presupposition). In both references, none of the intermediate tasks focus on pragmatics so while the references support the general conclusion that a carefully chosen intermediate task can help performance on downstream tasks, we also shared 4 other references in our Related Work section that support the same general conclusion as this is not the main point we are trying to emphasize in the paper. Specifically, since we’re interested in the performance of NLI on presupposition, we are exploring this unique linguistic angle that ties various notions of discourse and pragmatics (via the 3 chosen tasks) to presupposition and, as you rightly point out, show empirically that leveraging this linguistic link can help the performance of NLI models on cases of presupposition. In the camera-ready version, we'd be happy to add the two suggested references [1] and [2] to the Related Work section and highlight their contribution and how our work is unique and different from these two references.
>
> 2. Thank you for bringing up the question regarding test performance on MNLI. We did not include this analysis in our paper given the space limitation and because performance on MNLI is not central to the main point of the paper, namely, whether the linguistically motivated tasks can help with model performance on NLI cases involving presupposition. With extra space in the camera-ready version, we’d be happy to include these results and analyze any performance costs on MNLI.

---

### Meta-Review · Area_Chair_5XxL · 2023-09-16

**Recommendation:** 5

**Metareview:**

This work hypothesized that pre-finetuning NLI models with intermediate tasks related to discourse and pragmatics (namely coherence modeling, sarcasm detection, and discourse relation classification) would improve model performance on cases involving presupposition.

The paper provided empirical support for the authors' hypothesis, thus empirically establishing the connection between presupposition and these tasks. The reviewrs agreed that this was a useful result that could benefit future work. Overall, evaluation is solid with extensive ablation studies.

The rebuttal sufficiently addressed all of Reviewer joCE's concerns, particularly regarding the label randomization experiment and the use of the Vanilla setup as an informative baseline.

It's clear from Reviewer PE3F's response to the rebuttal that his/her concerns remained, specifically regarding the contribution of this paper in light of related work. I think the authors provided a reasonable response to PE3F's response and am personally satisified with their response. I view PE3F's concerns more as suggestions for improvements than serious flaws, and hope that the authors can take them into account when revising their paper.

---

### Decision · Program_Chairs · 2023-10-07

**Decision:**

Accept-Findings

**Comment:**

This work hypothesized that pre-finetuning NLI models with intermediate tasks related to discourse and pragmatics (namely coherence modeling, sarcasm detection, and discourse relation classification) would improve model performance on cases involving presupposition.

The paper provided empirical support for the authors' hypothesis, thus empirically establishing the connection between presupposition and these tasks. The reviewrs agreed that this was a useful result that could benefit future work. Overall, evaluation is solid with extensive ablation studies.

The rebuttal sufficiently addressed all of Reviewer joCE's concerns, particularly regarding the label randomization experiment and the use of the Vanilla setup as an informative baseline.

It's clear from Reviewer PE3F's response to the rebuttal that his/her concerns remained, specifically regarding the contribution of this paper in light of related work. I think the authors provided a reasonable response to PE3F's response and am personally satisified with their response. I view PE3F's concerns more as suggestions for improvements than serious flaws, and hope that the authors can take them into account when revising their paper.